# The Extract of *Gloiopeltis tenax* Enhances Myogenesis and Alleviates Dexamethasone-Induced Muscle Atrophy

**DOI:** 10.3390/ijms25126806

**Published:** 2024-06-20

**Authors:** Si-Hyung Kim, Young-Eun Leem, Hye Eun Park, Hae-In Jeong, Jihye Lee, Jong-Sun Kang

**Affiliations:** 1Department of Molecular Cell Biology, School of Medicine, Sungkyunkwan University, Suwon 16419, Republic of Korea; shk990929@skku.edu (S.-H.K.); leemyo@skku.edu (Y.-E.L.); 2Laboratories of Marine New Drugs, Redone Technologies Co., Ltd., Jangseong-gun 57247, Republic of Korea; phe@urc.kr (H.E.P.); jhl@urc.kr (H.-I.J.)

**Keywords:** *Gloiopeltis tenax*, red algae, muscle atrophy, PGC-1α, mitochondria

## Abstract

The decline in the function and mass of skeletal muscle during aging or other pathological conditions increases the incidence of aging-related secondary diseases, ultimately contributing to a decreased lifespan and quality of life. Much effort has been made to surmise the molecular mechanisms underlying muscle atrophy and develop tools for improving muscle function. Enhancing mitochondrial function is considered critical for increasing muscle function and health. This study is aimed at evaluating the effect of an aqueous extract of *Gloiopeltis tenax* (GTAE) on myogenesis and muscle atrophy caused by dexamethasone (DEX). The GTAE promoted myogenic differentiation, accompanied by an increase in peroxisome proliferator-activated receptor γ coactivator α (PGC-1α) expression and mitochondrial content in myoblast cell culture. In addition, the GTAE alleviated the DEX-mediated myotube atrophy that is attributable to the Akt-mediated inhibition of the Atrogin/MuRF1 pathway. Furthermore, an in vivo study using a DEX-induced muscle atrophy mouse model demonstrated the efficacy of GTAE in protecting muscles from atrophy and enhancing mitochondrial biogenesis and function, even under conditions of atrophy. Taken together, this study suggests that the GTAE shows propitious potential as a nutraceutical for enhancing muscle function and preventing muscle wasting.

## 1. Introduction

Sarcopenia is an aging-associated disease accompanied by muscle atrophy, characterized by increscent loss of muscle mass, strength, and function [1]. Decreased muscle mass and strength have adverse effects on individuals’ overall health and life quality, attributable to decreased mobility, balance issues, an increased risk of falls and fractures, and reduced functional independence [2,3]. Although the exact triggers of sarcopenia are still under investigation, hormonal changes, reduced physical activity, malnutrition, and certain chronic diseases appear to be important underlying factors in sarcopenia [4]. Furthermore, these conditions are highly linked to mitochondrial dysfunction, elevated oxidative stress, and reduced muscle regenerative capacity [5,6]. Physical exercise is commonly regarded as the most effective solution for muscle atrophy and weakness. However, there is a limitation in applying physical exercise to the elderly and patients who have difficulty with movement. Thus, much attention is paid to developing exercise-mimicking agents to intervene in sarcopenia [7,8].

Exercise leads to an increase in muscle strength and function, which are closely correlated with mitochondrial biogenesis and function [5,9]. PGC-1α is a master regulator of exercise’s effects, controlling the expression of genes related to energy metabolism and mitochondrial biogenesis. Aged muscle cells exhibit downregulation of PGC-1α transcription, which results in mitochondrial dysfunction [10]. In contrast, continuously exercised muscles showed a higher level of PGC-1α transcription [11]. Based on this, many groups have tried to search for muscle-enhancing drugs that activate mitochondrial metabolism by enhancing PGC-1α transcription in muscle cells.

Red algae are usually found in intertidal and subtidal regions along the coastlines of various continents, including North America and Asia. Some species of red algae, such as *Gloiopeltis tenax* (*G. tenax*), have been traditionally used for medicinal and therapeutic purposes. *G. tenax* is a one-year seaweed belonging to the red algae class. Specifically, it is a filamentous seaweed with stems, known to be distributed in various regions of South Korea. Recent studies have reported the suppressive effects of *G. tenax* extracts on cancer, chronic inflammation, and the generation of reactive oxygen species [12,13,14]. Oxidative stress and low-grade chronic inflammation have a strong association with the onset and development of muscle atrophy. This fact encouraged us to investigate the effect of *G. tenax* extracts on myoblast differentiation and muscle atrophy.

In the current study, *G. tenax* aqueous extract (GTAE) promoted myogenic differentiation and mitochondrial biogenesis, and showed significant protection against atrophy in myoblasts. In addition, the protective effect of GTAE was verified in a mouse model of DEX-induced muscle atrophy. The GTAE demonstrated a promising effect in protecting against muscle atrophy and enhancing mitochondrial metabolism in mice. Taken together, we demonstrate that GTAE could serve as an effective natural dietary supplement or therapeutic material for treating muscle atrophy.

## 2. Results

### 2.1. G. tenax Aqueous Extract Promotes PGC-1α Activity and Enhances Myogenic Differentiation

We utilized a PGC-1α-luciferase (Luc) construct to investigate the myogenic effects of extracts obtained under four distinct conditions of *G. tenax* (GTEH, GTER, GTAE, and GTEU) (Figure 1a). C2C12 myoblasts were transfected with PGC-1α-Luc, and 24 h later, they were treated with either DMSO or 1 µg/mL of four individual *G. tenax* extracts, followed by myogenic differentiation induction for an additional 24 h. The luciferase assay results showed that, among the extracts, the *G. tenax* aqueous extract (GTAE) exhibited approximately a 2-fold higher PGC-1α reporter activity level than the control. To assess whether the high PGC-1α reporter activity is linked to a high myogenic effect, we measured the levels of early myogenic markers, such as *eMHC* and *MyoG*, by conducting qRT-PCR on C2C12 cells that were induced with myogenic differentiation after treatment with the individual extracts (Figure 1b). Consistent with the luciferase assay data, the GTAE-treated cells exhibited the highest expression levels of *eMHC* and *MyoG* compared to the other tested extracts. Based on the results, GTAE was selected as an effective candidate for inducing myogenesis through the activation of PGC-1α.

With GTAE, we initially decided to examine its cytotoxicity by performing an MTT assay in the range of 0.1, 1, and 10 μg/mL (Figure 1c). We found that none of the amounts of GTAE affected cell viability. Then, C2C12 cells were cultured until differentiation day 0 (D0) and treated with various concentrations of GTAE. After 24 h of myogenic induction, the effect of GTAE on myogenesis was evaluated via qRT-PCR (Figure 1d). The results showed that the promotion of myogenic differentiation by GTAE was dose-dependent. As the amount of GTAE increased, the expression of myogenic markers also increased. The myogenic enhancing effect of GTAE was also confirmed by MHC staining in cells treated with GTAE and induced to differentiate until D3 (Figure 1e–g). As the amount increased, GTAE exhibited an increase in both the diameter of myotubes and the number of nuclei in MHC-positive myotubes. Consistent with this result, the protein level of MHC rose as the amount of GTAE increased (Figure 1h,i and Appendix A). Subsequently, we identified the effect of GTAE on protein synthesis in myoblasts by determining the activity of the PI3K/Akt/mTOR signaling pathway after treatment with GTAE (Figure 1j,k and Appendix A). As the amount of GTAE increased, the phosphorylation levels of Akt and mTOR increased proportionally. Therefore, these results indicate that the GTAE extract of *G. tenax* elicits a strong PGC-1α activity and exerts a myogenic effect through the Akt/mTOR signaling pathway.

### 2.2. Identification of Compounds of GTAE by LC-MS Analysis

The identification of three compounds in GTAE was achieved through the analysis of liquid chromatography–mass spectrometry (LC-MS) and NMR spectroscopic data. Specifically, 4-Hydroxy-*N*-methylproline, floridoside, and shinorine were detected by comparing their mass data (Appendix A). Additionally, the isolation of compounds from GTAE using HPLC-UV was carried out (Appendix A), and their chemical structures were elucidated by comparing their ^1^H NMR spectra with previously reported data (Appendix A) [15,16,17].

### 2.3. GTAE Upregulates Mitochondrial Biogenesis and Function in Myoblasts

Mitochondria, the energy-producing organelles, play a crucial role in both myoblast differentiation and muscle function [9]. Based on this, we decided to evaluate the effect of GTAE on mitochondrial mass and function. Initially, we isolated mitochondrial DNA (mtDNA) from C2C12 cells that were incubated with a vehicle or 10 μg/mL GTAE for 24 h, and then estimated the ratio of mtDNA to nuclear DNA (nDNA) (Figure 2a). The result showed that GTAE elevated the level of mtDNA compared to the control, indicating that GTAE contributes to an increase in mitochondrial mass in myoblasts. Next, we assessed the influence of GTAE on mitochondrial functions during myogenic differentiation. To achieve this, C2C12 myoblasts were differentiated in the presence of a vehicle or GTAE at the indicated concentrations for 3 days (D3). Then, immunoblotting was performed for PGC-1α and oxidative phosphorylation (OXPHOS) proteins (Figure 2b,c and Appendix A). The result exhibited that as the concentration of GTAE increased, the expression levels of PGC-1α and OXPHOS proteins increased. Additionally, the mRNA levels of mitochondria-related genes were also augmented in GTAE-treated myoblasts compared to the vehicle-treated cells at D3 (Figure 2d). To confirm these results, we administered JC-1 dye to myoblasts treated with either a vehicle or GTAE for 24 h and then evaluated the mitochondrial membrane potential (Figure 2e,f). The GTAE treatment resulted in a higher ratio of red aggregates to green monomers compared to the vehicle treatment, indicating that GTAE leads to a higher mitochondrial membrane potential in myoblasts. Therefore, all these data suggest that the GTAE extract has a positive effect on mitochondrial biogenesis and activity during myogenic differentiation.

### 2.4. GTAE Prevents the Reduction in Myotube Size in an In Vitro Atrophy Model

To verify the protective effect of GTAE against muscle atrophy, we decided to examine the effect of GTAE on DEX-triggered myotube atrophy. C2C12 cells were differentiated for 3 days to form myotubes and then treated with DEX and/or 10 μg/mL GTAE (Figure 3a). After 24 h, the cells were subjected to MHC staining, and the myotube diameters and the number of nuclei in MHC-positive myotubes were estimated (Figure 3b–d). The DEX treatment caused atrophy in myotubes, as indicated by the decline in both myotube diameter and the number of multinucleated myotubes compared to the control cells. However, GTAE prevented the DEX-induced atrophy of myotubes, as evidenced by the increase in multinucleated myotube size compared to DEX-treated cells. To validate this data, we performed qRT-PCR to verify the mRNA levels of atrophy marker genes, such as Atrogin-1/Fbxo32 and Murf1/Trim63 (Figure 3e). The elevated levels of atrophy genes observed in DEX-treated cells were substantially reduced when cells were treated with GTAE in addition to DEX. Similarly, their protein expression was also reduced in GTAE-treated, DEX-induced atrophic cells compared to DEX-treated cells (Figure 3f,g and Appendix A). Consequently, the administration of GTAE in the condition of DEX-induced atrophy restored the level of phosphorylated Akt, which is linked to protein synthesis, along with MHC expression. All of this data suggests that GTAE attenuates DEX-induced atrophy in an in vitro condition through the regulation of muscle protein degradation.

### 2.5. GTAE Alleviates DEX-Triggered Muscle Atrophy in Mice

The positive effects of GTAE on myogenic differentiation and alleviation of atrophy inspired us to evaluate its protective effect against muscle atrophy by using a DEX-induced muscle atrophy mouse model. Eight-week-old mice were orally administered a vehicle or GTAE daily for 7 days, followed by daily I.P. injection of DEX, along with the administration of a vehicle or GTAE for an additional 10 days. Prior to the sacrifice, physical activity tests were performed to measure the capability of muscle strength (Figure 4a). The DEX (=DEX-Veh) group exhibited significant body weight loss compared to the Con (=Con-Veh) group, despite consuming a relatively high amount of food (Figure 4b,c). Interestingly, the body weight of the DEX-GTAE group was comparable to that of the Con group. To assess the effect of GTAE on muscle function, we conducted grip strength and treadmill tests (Figure 4d,e). The results showed that the DEX group exhibited a significant reduction in both the isometric strength of forelimb muscles and aerobic endurance compared to the Con group, indicating successful DEX-triggered muscle atrophy in mice. The oral administration of GTAE rescued muscle strength and endurance in the DEX-GTAE groups when compared to the DEX group. When we analyzed and compared the weight of muscle tissues, the DEX-induced atrophy led to a decrease in the weight of the extensor digitorum longus (EDL) muscles (Figure 4f). However, the administration of GTAE slightly restored the muscle weight, even in the condition of atrophy caused by DEX. Meanwhile, the difference in weight of white adipose tissue (WAT) was hardly noticeable in all groups of mice (Figure 4g). Then, we examined the cross-sectional area of myofibers in EDL muscles by conducting immunofluorescence staining against laminin and MyhIIa, as well as laminin and MyhIIb (Figure 4h,i). The injection of DEX led to a decrease in the size of both types of muscle fibers, but the administration of GTAE to DEX-injected mice recovered the muscle fiber diameter. To confirm the rescue effect, we determined the alteration of gene expression associated with atrophy in gastrocnemius (GAS) muscles by performing qRT-PCR (Figure 4j). Consistent with the staining results, the DEX-GTAE group showed a decrease in the expression of *Atrogin-1* and *MuRF-1* compared to the DEX group, indicating the alleviation of muscle atrophy caused by DEX. Therefore, all of these data suggest that GTAE alleviates muscle atrophy and contributes to an increase in muscle function and mass in mice with DEX-induced atrophy.

### 2.6. GTAE Enhances Muscle Mitochondrial Metabolism in DEX-Induced Muscle Atrophy

To assess the effect of GTAE on mitochondrial biogenesis, mtDNA was extracted from the tibialis anterior (TA) muscles of each group and quantified for the mtDNA/nDNA ratio (Figure 5a). The DEX group exhibited a decrease in mtDNA levels compared to the Con group. However, the DEX-GTAE group showed a recovery of the mtDNA/nDNA ratio compared to the DEX group. This result prompted us to compare the levels of total OXPHOS complex proteins between the groups in order to assess mitochondrial function (Figure 5b,c and Appendix A). The results showed that DEX-induced atrophy caused a reduction in the expression of the total OXPHOS complex, and this reduction was rescued by GTAE administration. Then, the oxidative capacity in the EDL muscles of each group was evaluated by performing immunohistochemistry for SDH or NADH (Figure 5d,e). The DEX-induced muscle atrophy reduced the proportion of myofibers with strong (dark) and mild (intermediate) activities for SDH and NADH compared to the Con group. However, the administration of GTAE elevated the oxidative capacity of the muscles, even under the atrophy condition. The qRT-PCR analysis results for the expression levels of mitochondrial genes were consistent with data from the prior analyses, showing that GTAE led to the recovery of mitochondrial gene expression under the DEX-treated condition (Figure 5f). Taken together, these results suggest that GTAE mitigates the decline in mitochondrial metabolism caused by muscle atrophy and has a protective effect against muscle atrophy.

## 3. Discussion

Seaweeds contain bioactive compounds that are not found in terrestrial food sources and are receiving attention for their efficacy in treating various diseases such as cardiovascular diseases, metabolic diseases, diabetes, and cancers [18,19,20]. In the current study, we were interested in red algae, which are the most diverse group of seaweeds and are abundant in the intertidal and subtidal zones. They contain high-quality proteins, making them alternative protein sources [21]. *Gloiopeltis*, a genus of red algae, contains seven species, among which *G. furcata* and *G. tenax* are edible. The medicinal effects of red algae have predominantly been studied with *G. furcata*, which has been identified as having antioxidant, antitumor, and antidiabetic effects [22,23,24,25,26]. A recent study reported that an ethanol extract of *G. furcata*, as well as its effective component Docosahexaenoic acid (DHA), improved physical performance in mice [27]. However, they mainly focused on the effects of physical activity without molecular analysis on muscle protein expression, atrophy-related gene expression, muscle differentiation, and mitochondrial gene expression.

In the current study, we explored the potential benefits of the extract from *G. tenax*. We initially prepared several extracts of *G. tenax* through ethanol extraction or water extraction. Three different ethanol extracts (GTEH, GTER, and GTEU) were tested for their effects on *PGC-1α* transcriptional activation and myogenic gene expressions. Their activation effects were found to be lower than that of the water-soluble GTAE. The aqueous extract, GTAE, showed an increase in myogenic differentiation and mitochondrial metabolism, as well as an attenuation of muscle atrophy, both in vitro and in vivo. Thus, it is conceivable that the effects of the *G. tenax* aqueous extract on physical performance are mediated by PGC-1α and mitochondrial enhancement. Furthermore, an effective component of GTAE remains to be discovered in the future. Although it requires identifying the constituents of GTAE to clarify its atrophy-protective effect, we cannot rule out that taurine could be one of the constituents of GTAE. Taurine is water-soluble and has been found to be present in *G. tenax* in high concentrations [28]. It is known to play a positive role in skeletal muscle function by being involved in osmotic homeostasis, protein stability, oxidative stress, mitochondrial protein synthesis, and calcium homeostasis [29,30].

PGC-1α is a master regulator for mitochondrial homeostasis and antioxidant protection [31,32]. In skeletal muscles, the transcription of PGC-1α is regulated by MEF2, CREB, AMPK, and p38MAPK [33]. Also, there are several isoforms of PGC-1α that are produced by alternative splicing. Among them, PGC-1α-b is known to be an exercise-associated isoform [34,35]. Furthermore, PGC-1α can be modulated through post-translational modifications, such as ubiquitination, phosphorylation, and acetylation [33]. As ATP production organelles, mitochondria are responsible for regulating the metabolic and functional status of skeletal muscle [36,37]. Considering that PGC-1α is closely associated with the regulation of the content and function of mitochondria, it is obvious that the increase in the expression and activation of PGC-1α facilitates muscle function. Practically, it was discovered that the level of PGC-1α increases in continuously exercised muscles, where PGC-1α is involved in antioxidant protection, mitochondrial biogenesis, and fiber type switching to Type I [11]. Meanwhile, in aged muscle cells, the transcription of PGC-1α decreases, resulting in impaired mitochondrial homeostasis and mitochondrial dysfunction [10]. Furthermore, PGC-1α is also involved in suppressing the loss of muscle mass by inhibiting FoxO3 activity [38]. FoxO3 is a key regulator of the expression of atrophy-related ubiquitin ligases, such as Atrogin-1, leading to muscle protein degradation. PGC-1α restricts the recruitment of FoxO3 to a FoxO3 response element in the *Atrogin-1* promoter, and transfection of PGC-1α into TA muscles prevents muscle fiber atrophy stimulated by FoxO3. Under the conditions of atrophy induced by denervation or fasting, transgenic mice overexpressing PGC-1α exhibit alleviation in the reduction in muscle fiber size, decreased expression of Atrogin-1 and MuRF1, and increased expression of energy-metabolism-related genes compared to wild-type mice.

## 4. Materials and Methods

### 4.1. Materials and Preparation of GTAE

*G. tenax* was collected in Wando-gun (Jeollanam-do, Republic of Korea). The collected materials underwent three wash cycles followed by air-drying at room temperature for 24 h. The dried seaweed underwent extraction utilizing four distinct methods: ethanol extraction at 60 °C for one hour (GTEH), ethanol extraction at room temperature for one hour (GTER), water extraction at 60 °C for one hour (GTAE), and ultrasonic-assisted ethanol extraction at room temperature for one hour (GTEU). Following extraction, the filtrate was obtained through non-fluorescent cotton filtration, and subsequent removal of distilled water was achieved via a rotary evaporator (WEV-1001V; Daihan Scientific, Seoul, Republic of Korea) at 36 °C.

### 4.2. General Experimental

Nuclear magnetic resonance (NMR) spectra were recorded at 400 MHz in D_2_O and CD_3_OD using Varian Inova NMR spectrometers (Varian, Inc., Palo Alto, CA, USA). LC-MS was conducted with an Agilent Technologies 1260 (Agilent Technologies, Inc., Santa Clara, CA, USA) and a Waters Micromass ZQ LC-MS system (Waters Corporation, Milford, MA, USA), utilizing a hydrophilic interaction liquid chromatography (HILIC) column (Phenomenex Luna HILIC, 150 mm × 2 mm, 3 µm; Phenomenex, Torrance, CA, USA) at a flow rate of 0.2 mL/min. The GTAE was purified using a Waters 1525 binary high-performance liquid chromatography (HPLC) pump equipped with a HILIC column (Phenomenex Luna HILIC, 250 mm × 10 mm, 5 µm) at a flow rate of 1.0 mL/min.

### 4.3. LC-MS Analysis and Isolation of Compounds

For the sample preparation, 2 mg of GTAE was placed in a 1.5 mL Eppendorf tube, mixed with 1 mL of water, filtered through a 0.45 µm filter, and injected into the LC-MS system. The binary gradient elution system used consisted of 90% acetonitrile (ACN) with 5 mM ammonium formate (NH_4_HCO_2_) and 50% ACN with 5 mM NH_4_HCO_2_ as eluents. The gradient for LC-MS data acquisition was as follows: 0.0–1.5 min, 5% 50% ACN/5 mM NH_4_HCO_2_; 1.6–10.0 min, 5–50% 50% ACN/5 mM NH_4_HCO_2_; 10.1–12.5 min, 50% 50% ACN/5 mM NH_4_HCO_2_.

The GTAE (1 g) was further purified by HPLC using an isocratic solvent system at a column temperature of 40 °C. The isocratic separation employed a mixture of 90% ACN with 5 mM NH_4_HCO_2_ and 50% ACN with 5 mM NH_4_HCO_2_ (80:20, *v*/*v*), resulting in the isolation of 4-hydroxy-N-methylproline (4.1 mg), floridoside (41.0 mg), and shinorine (18.4 mg).

### 4.4. Cell Culture and Differentiation

C2C12 myoblasts were purchased from ATCC (Manassas, VA, USA) and cultured as previously described [39]. Briefly, C2C12 cells were grown in growth medium (GM; Dulbecco’s Modified Eagle Medium high glucose (DMEM; Thermo Scientific, Waltham, MA, USA) containing 15% fetal bovine serum (FBS) and 1× penicillin/streptomycin) at 37 °C, 5% CO_2_. To induce myogenic differentiation, cells at high confluence (cell confluency of almost 80–90%) were switched to differentiation medium (DM; DMEM containing 2% horse serum and 1× penicillin/streptomycin). To examine the effect of GTAE on DEX-induced atrophy, C2C12 cells were treated with 100 μM DEX and the indicated amount of GTAE on differentiation day 3 (D3). Subsequently, they were allowed to differentiate for an additional day and were then examined for myotube formation.

### 4.5. Animal Studies and Physical Activity Tests

C57Bl/6 male mice were obtained from Orient-Bio (Seongnam, Korea) and were housed 23 °C under a 12:12 h light/dark cycle with ad libitum access to food and water. Eight-week-old male mice were orally administered a daily dose of 8 mg/kg of either vehicle or GTAE for one week prior to the intraperitoneal (I.P.) injection of DEX (20 mg/kg). The administration of vehicle/GTAE and the I.P. injection of DEX continued for an additional 10 days before the mice were sacrificed. The physical activity tests were conducted on day 7 and day 8 after the injection. After fasting for 16 h, the mice were sacrificed, and their muscles were harvested following the final administration of either a vehicle or GTAE. The grip strength test was performed following the procedure described in the previous study [40]. Briefly, mice were allowed to grab the grid with their forelimbs and were then gently pulled with a consistent force until the limbs were detached from the grid. The maximal strength was recorded during a blind test. For the treadmill test, mice were initially acclimated to the treadmill (Columbus Instruments Exer-6M Treadmill, Columbus, OH, USA) by running at speeds of 5 m/min for 10 min, 13 m/min for 4 min, and 17 m/min for 1 min, while maintaining a 10% slope. The following day, the treadmill test was performed with a starting speed of 8 m/min, followed by an increase of 1 m/min at 2 min intervals. This running was conducted until the mice were exhausted. All animal experiments were approved by the Institutional Animal Care and Use Committee (IACUC) of Sungkyunkwan University School of Medicine (SUSM) and complied with the animal experimenting guidelines of the SUSM ethics committee.

### 4.6. Cell Viability Assay and Luciferase Assay

Cell viability was determined by performing the MTT assay. Briefly, C2C12 cells were seeded and cultured in a 96-well dish with a density of 5 × 10^4^ cells/well for 24 h. They were then treated with the indicated concentration of vehicle/GTAE for 24 h. To measure cell viability, the cells were incubated with MTT solution at 37 °C for 4 h. After dissolving the MTT formazan in DMSO, the absorbance was measured at 540 nm. To perform the luciferase assay, C2C12 cells were transfected with a PGC-1α-luciferase construct and a β-galactosidase plasmid using Lipofectamine 2000 (Invitrogen, Waltham, MA, USA) according to the manufacturer’s instructions. The transfectants were induced for myogenic differentiation until D1, and then the cells were subjected to analysis of luciferase activity, as instructed by the manufacturer (Promega, Waltham, MA, USA).

### 4.7. Western Blot Analysis

Cells were lysed using a lysis buffer consisting of 50 mM Tris-HCl (pH 7.4), 1.5 mM MgCl_2_, 150 mM NaCl, 1 mM EGTA, 1% Triton X-100, and a complete protease inhibitor cocktail. After running SDS-PAGE and transferring to the membrane, the incubation with primary and secondary antibodies was performed. The antibodies used in this study are listed in Appendix A. For loading control, either GAPDH, HSP90, or β-Actin was used due to the cross-reactivity of antibodies or the differences in molecular weights of the proteins during the reblotting process. To quantify the protein levels, each band intensity was determined with ImageJ software (https://imagej.net/ij/index.html, NIH, Bethesda, MD, USA) and was normalized to the corresponding loading control or total protein level.

### 4.8. Immunofluorescence Microscopy and Cryosection

To perform immunofluorescence staining, the cells were fixed with 4% paraformaldehyde, permeabilized with 0.2% Triton X-100 in PBS, and then blocked with 5% goat serum in PBS. For MHC staining, the cells were incubated with anti-MHC antibodies, followed by incubation with the secondary antibody and counterstaining with DAPI. The isolated muscles were embedded in Tissue-Tek OCT Compound (Sakura Finetek, Nagano, Japan) and cryosectioned to 7 μm thickness using a cryomicrotome. After fixation and permeabilization, cryosections were incubated with primary antibodies to Myosin heavy chain (Myh) type IIA, Myh type IIB (DSHB), or laminin (Abcam, Cambridge, UK) antibodies, followed by secondary antibody incubation. The fluorescence images were captured and processed using a LSM-710 confocal microscope system (Carl Zeiss, Baden-Wurttemberg, Germany) and a Nikon ECLIPSE TE-2000U (Nikon, Tokyo, Japan) with NIS-Elements BR 4.30.01 software and ImageJ software.

### 4.9. Real-Time Quantitative RT-PCR

Total RNA was isolated from muscles by homogenizing the tissues using 1 mm and 2 mm glass beads (Glastechnique Mfg, Wertheim, Germany), easyBLUE total RNA extraction solution (iNtRON Biotechnology, Seongnam, Republic of Korea), and the FastPrepR-24 (MP Biomedicals, Santa Ana, CA, USA). After isolating total RNAs, cDNA was synthesized using the PrimeScript RT reagent kit (TaKaRa, Shiga, Japan) in accordance with the manufacturer’s instructions. Real-time qRT-PCR was performed using the SYBR Premix ExTaq kit (TaKaRa) and Thermal Cycler Dice real-time system (TaKaRa). The primers used in this study are indicated in Appendix A.

### 4.10. Mitochondrial Analysis

To assess mitochondrial DNA contents, total DNA was extracted using the DNeasy Blood and Tissue kit (QIAGEN, Venlo, The Netherlands) and the amount of mitochondrial DNA was quantified by the ratio of Mt-co1 to β-Actin by quantitative PCR. qPCR was performed using the TaKaRa Taq DNA polyerase kit (TaKaRa) and SimpliAmp thermal cycler (Applied Biosystems, Waltham, MA, USA). The primers are indicated in Appendix A. For JC-1 staining, cells were stained with 2 μg/mL JC-1 dyes (MP 03168, Invitrogen) for 30 min, washed with Live Cell Imaging Solution (A14291DJ, Invitrogen), and then imaged using a confocal microscope system (Carl Zeiss). For the analysis of NADH dehydrogenase activity, the dried sections were incubated in a solution containing 0.9 mM NADH and 1.5 mM Nitro blue tetrazolium (NBT; Sigma-Aldrich, St. Louis and Burlington, MA, USA) in 3.5 mM phosphate buffer (pH 7.4) for 30 min. To measure the succinate dehydrogenase (SDH) activity, the sections were reacted with 50 μM sodium succinate and 0.3 mM Nitro blue tetrazolium in 114 mM phosphate buffer containing K-EGTA (Sigma-Aldrich) for 1 hr. All images were captured using a Nikon ECLIPSE TE-2000U (Nikon) and were analyzed with NIS-Elements F software and ImageJ software.

### 4.11. Statistical Analysis

Values were expressed as mean ± SD or ± SEM, as indicated in the figure legends. The statistical significance was calculated using an unpaired, two-tailed Student’s *t*-test. Differences were considered statistically significant at or below * *p* < 0.05, ** *p* < 0.01, and *** *p* < 0.001.

## 5. Conclusions

We provide evidence of the potential benefits of *G. tenax* extract in alleviating skeletal muscle atrophy through the activation of PGC-1α gene expression. Thus, GTAE could be a promising therapeutic substance for managing muscle atrophy and promoting muscle regeneration.

## Figures and Tables

**Figure 1 ijms-25-06806-f001:**
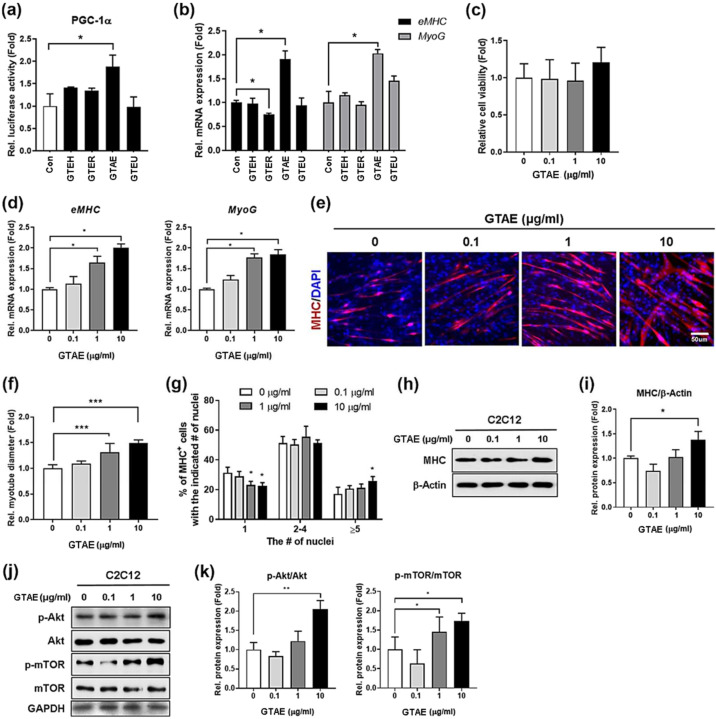
The aqueous extract of *G. tenax* (GTAE) promotes PGC-1α activity and enhances myogenic differentiation. (**a**) Each extract (GTEH, GTER, GTAE, and GTEU) of *G. tenax* (1 μg/mL) was tested for PGC-1α activation using a PGC-1α-Luc construct in C2C12 cells. (**b**) qRT-PCR analysis for the expression of early muscle differentiation markers (*eMHC* and *MyoG*) in C2C12 cells treated with individual extracts (1 μg/mL) of *G. tenax*. (**c**) MTT assay was performed to determine the cell viability in C2C12 cells treated with 0.1, 1, or 10 μg/mL GTAE. (**d**) qRT-PCR analysis for the expressions of *eMHC* and *MyoG* in C2C12 cells that were treated with a vehicle, 0.1, 1, or 10 μg/mL GTAE at D0 and differentiated for 24 h. (**e**) Immunostaining for MHC in C2C12 cells that were treated with a vehicle, 0.1, 1, or 10 μg/mL GTAE at D0 and were induced for myogenic differentiation until D3. Scale bar = 50 μm. (**f**) Quantification for the diameter of MHC-positive myotubes shown in panel e (n = 3). (**g**) The percentages of MHC-positive myotubes containing the indicated number of nuclei, as shown in panel e (n = 3). (**h**) Western blot analysis for the expression of MHC in C2C12 cells that were treated with a vehicle, 0.1, 1, or 10 μg/mL GTAE at D0 and then induced for myogenic differentiation until D3. β-Actin was used as a loading control. (**i**) Quantification of the relative protein levels of MHC, shown in panel h (n = 3). (**j**) Western blot analysis for the levels of p-Akt, Akt, p-mTOR, and mTOR in C2C12 cells that were treated with a vehicle, 0.1, 1, or 10 μg/mL GTAE at D0 and were then induced for differentiation for an additional 3 days. GAPDH was used as a loading control. (**k**) The relative signal intensity of p-Akt and p-mTOR shown in panel j was quantified and normalized to that of total Akt and mTOR, respectively (n = 3). A one-way ANOVA analysis with a Tukey post hoc test was utilized to determine the statistical significance. Data is represented as the mean ±SD or ± SEM. Asterisks indicate the significant difference from the control. * *p* < 0.05, ** *p* < 0.01, and *** *p* < 0.001.

**Figure 2 ijms-25-06806-f002:**
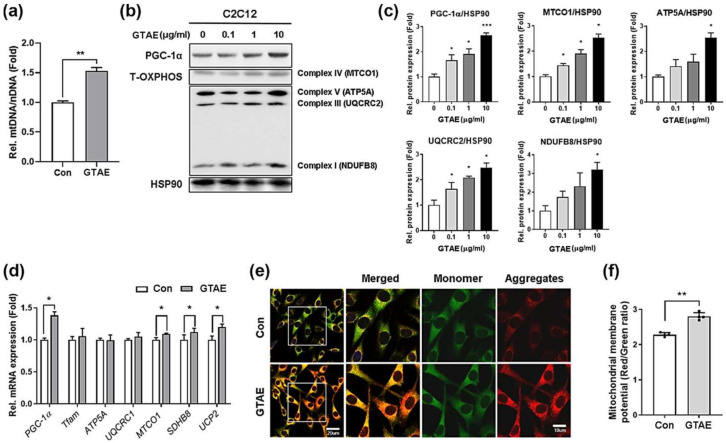
GTAE upregulates mitochondrial biogenesis and function in myoblasts. (**a**) qPCR analysis for measuring mtDNA/nDNA ratio in C2C12 cells that were treated with either a vehicle or 10 μg/mL GTAE for 24 h in the GM. (**b**) Western blot analysis for the expressions of PGC-1α and total OXPHOS complex proteins in C2C12 cells that were treated with a vehicle, 0.1, 1, or 10 μg/mL GTAE at D0 and then differentiated until D3. HSP90 was used as a loading control. (**c**) Quantification of the levels of PGC-1α and total OXPHOS proteins shown in panel b (n = 3). (**d**) qRT-PCR analysis for the expressions of PGC-1α and mitochondria-related genes in C2C12 cells that were treated with either a vehicle or 10 μg/mL GTAE at D0 and then differentiated for an additional 3 days. L-32 was selected as an endogenous control. (**e**) JC-1 staining was performed on C2C12 cells treated with either a vehicle or 10 μg/mL GTAE for 24 h in the GM. The images of the white boxes in the left row panels were enlarged on the right. Scale bar = 20 μm; 10 μm (inset). (**f**) Quantification of JC-1 staining shown in panel e. The relative intensity of JC-1 aggregates (red) to JC-1 monomers (green) was quantified. A one-way ANOVA analysis with a Tukey post hoc test was used to determine the statistical significance. Data is expressed as the mean ±SD or ± SEM. Asterisks indicate the significant difference from the control. * *p* < 0.05, ** *p* < 0.01, and *** *p* < 0.001.

**Figure 3 ijms-25-06806-f003:**
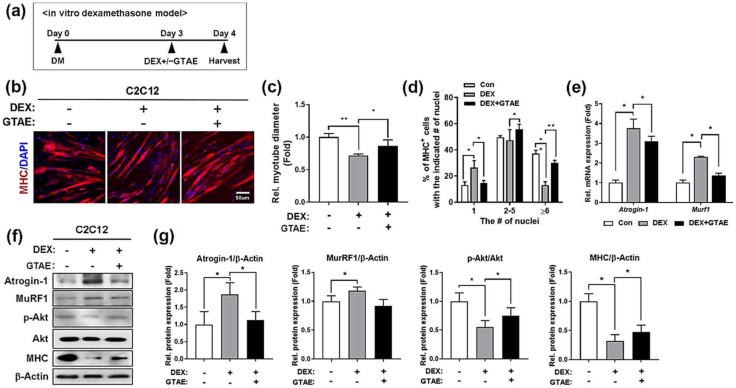
GTAE alleviates the reduction in myotube size in an in vitro atrophy model. (**a**) The procedure diagram depicts DEX-induced atrophy in C2C12 cells. C2C12 cells were differentiated for 3 days and were then treated with a vehicle or 10 μg/mL GTAE and/or DEX (100 μM) for an additional 1 day in the DM. (**b**) Immunostaining for MHC (red) was performed in C2C12 cells that were differentiated for 3 days in the DM. Afterward, they were treated with a vehicle or 10 μg/mL GTAE and/or DEX for an additional day. Nuclei were counter-stained with DAPI (blue). Scale bar = 50 μm. (**c**) Quantification of the relative diameter of the MHC-positive myotubes shown in panel b (n = 5). (**d**) Quantification of MHC-positive myotubes containing the indicated number of nuclei, as shown in panel b (n = 5). (**e**) qRT-PCR for *Atrogin-1* and *Murf1* in C2C12 cells treated with a vehicle or 10 μg/mL GTAE and/or DEX (100 μM) for 24 h in the DM. (**f**) Western blot analysis for the levels of MHC, MuRF1, Atrogin-1, p-Akt, and Akt in C2C12 cells treated with a vehicle or 10 μg/mL GTAE and/or DEX (100 μM). β-Actin was used as a loading control. (**g**) The band intensities of MuRF1, Atrogin-1, and MHC shown in panel f were quantified, and each value was normalized to that of β-Actin. The signal intensity of p-Akt was measured and then normalized to total Akt. An unpaired two-tailed Student’s *t*-test and ANOVA analysis with Tukey post hoc test were both utilized to determine the statistical significance. Data is presented as the mean ± SD or ± SEM. Asterisks indicate a significant difference from the control. * *p* < 0.05 and ** *p* < 0.01.

**Figure 4 ijms-25-06806-f004:**
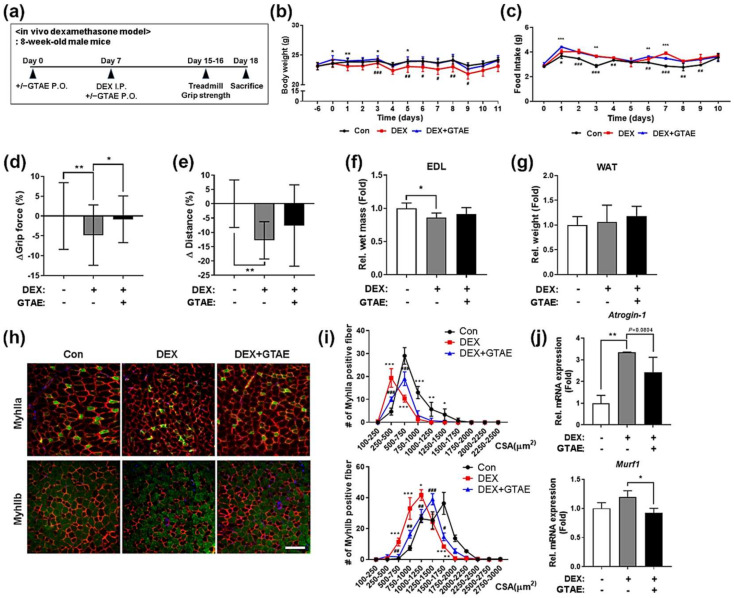
GTAE alleviates DEX-induced muscle atrophy in mice. (**a**) The procedure diagram depicts the in vivo study of DEX-induced muscle atrophy. Eight-week-old male mice were administered a vehicle or GTAE (8 mg/kg) by P.O. for 7 days. Then, a P.O. of a vehicle or GTAE and/or an I.P. injection of DEX (20 mg/kg) was conducted for an additional 10 days. For the physical activity tests, the treadmill test and the grip strength test were conducted 2–3 days prior to the sacrifice. (**b**) The changes in body weights of mice administered with a vehicle or GTAE and/or DEX (Con (n = 9), DEX (n = 9), DEX + GTAE (n = 10). (**c**) The changes in food intake of mice treated with a vehicle or GTAE and/or DEX (Con (n = 9), DEX (n = 9), DEX + GTAE (n = 10). (**d**) The difference in grip forces of mice administered with a vehicle or GTAE and/or DEX (Con (n = 9), DEX (n = 9), DEX + GTAE (n = 10). (**e**) The endurance activity was measured by the treadmill test with mice administered with a vehicle or GTAE and/or DEX (Con (n = 9), DEX (n = 9), DEX + GTAE (n = 10). (**f**) The wet weights of the EDL muscles from mice administered with a vehicle or SMGL and/or DEX (Con (n = 9), DEX (n = 9), DEX + GTAE (n = 10). (**g**) The wet weights of the WAT from mice administered with a vehicle or GTAE and/or DEX (Con (n = 9), DEX (n = 9), DEX + GTAE (n = 10). (**h**) Immunostaining for MyhIIa (green), MyhIIb (green), laminin (red), and DAPI (blue) was performed in the EDL muscles from mice administered with a vehicle or GTAE and/or DEX. Scale bar = 50 μm. (**i**) The quantification of the cross-sectional area of MyhIIa- or MyhIIb-positive myofibers shown in panel h (n = 3). (**j**) qRT-PCR for *Atrogin-1* and *Murf1* expression in the GAS muscles from mice treated with a vehicle or GTAE and/or DEX (n = 3). The two-way ANOVA analysis with Tukey post hoc test was used to determine statistical significance. Data is presented as the mean ± SD or ± SEM. Asterisks indicate a significant difference from the control. * *p* < 0.05, ** *p* < 0.01, and *** *p* < 0.001. # *p* < 0.05, ## *p* < 0.01, and ### *p* < 0.001.

**Figure 5 ijms-25-06806-f005:**
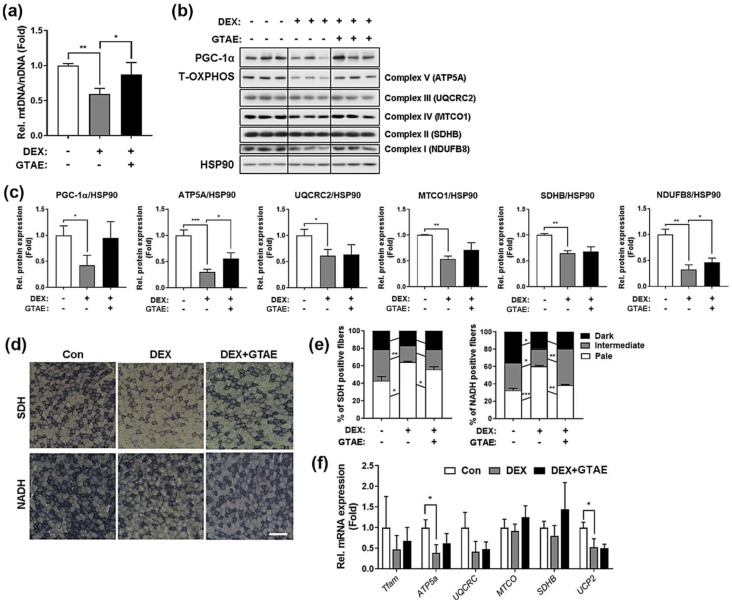
GTAE enhances muscle mitochondrial metabolism in DEX-induced muscle atrophy. (**a**) qPCR analysis for measuring the mtDNA/nDNA ratio of the TA muscles from mice administered with a vehicle or GTAE and/or DEX (n = 3). (**b**) Western blot analysis for the expression levels of PGC-1α and OXPHOS proteins in the TA muscles. HSP90 was used as a loading control. (**c**) Quantification of the levels of PGC-1α and OXPHOS proteins shown in panel b (n = 3). (**d**) Histochemical staining for SDH and NADH-TR enzymatic activities in the TA muscles. Scale bar = 50 μm. (**e**) The staining intensities of SDH and NADH-TR were quantified using three different grades (dark, intermediate, and pale) and plotted as percentiles (n = 3). (**f**) qRT-PCR analysis for the expression of mitochondria-related genes in the TA muscles from mice treated with a vehicle or GTAE and/or DEX (n = 3). The two-way ANOVA analysis with Tukey post hoc test was used to determine statistical significance. Data is presented as the mean ± SD or ± SEM. Asterisks indicate a significant difference from the control. * *p* < 0.05, ** *p* < 0.01, and *** *p* < 0.001.

## Data Availability

The data presented in this study are available in Appendix A.

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
