# Peer review of "The Extract of *Gloiopeltis tenax* Enhances Myogenesis and Alleviates Dexamethasone-Induced Muscle Atrophy"

_ijms, 2024, doi:10.3390/ijms25126806_

Round 1

Reviewer 1 Report

Comments and Suggestions for Authors

In the article: “The extract of Gloiopeltis tenax enhances myogenesis and alleviates dexamethasone-induced muscle atrophy”, the authors effect of the aqueous extract of Gloiopeltis tenax on myogenesis and muscle atrophy caused by dexamethasone. 

 We would like to invite the authors  to better clarify some points:

 1.       Please check the check punctuation and spaces;

2.       Page 1, lines 37-40, “Physical exercise is commonly regarded as the most effective solution for muscle atrophy and weakness. However, there is a limitation in applying physical exercise to the elderly and patients who have difficulty with movement. Thus, much attention is paid to developing exercise-mimicking agents to intervene in sarcopenia”. Relative reference is missing;

3.       Within the introduction the authors should briefly introduce the concept of muscle athropy, the following reference should be useful; Stellavato, A., Abate, L., Vassallo, V., Donniacuo, M., Rinaldi, B., & Schiraldi, C. (2020). An in vitro study to assess the effect of hyaluronan-based gels on muscle-derived cells: Highlighting a new perspective in regenerative medicine. PloS one, 15(8), e0236164. https://doi.org/10.1371/journal.pone.0236164;

4.       Figure 2B; scale bar are missing, please ass also information about the used magnification;

5.       Figure 2E, 3B, 4H and 5D; the same of Figure 2B;

6.       Why did you use housekeeping different for western blotting? Please, specify among the text the reasons;

7.       Figure 5B; western blotting illustration is not clear;

8.       It is not clear the origin of used cells, please, give more details;

9.       Please, insert specific details about antibodies codes;

10.   Please, try to better describe RNA extraction.

Comments on the Quality of English Language

In this manuscript minr mistakes based on spelling and/or editing are present.

Author Response

  1. Please check the check punctuation and spaces;

Response: As suggested by the reviewer, we carefully edited the manuscript for punctuation and spaces.

  1. Page 1, lines 37-40, “Physical exercise is commonly regarded as the most effective solution for muscle atrophy and weakness. However, there is a limitation in applying physical exercise to the elderly and patients who have difficulty with movement. Thus, much attention is paid to developing exercise-mimicking agents to intervene in sarcopenia”. Relative reference is missing;

Response: As advised by the reviewer, we added references after these sentences.

  1. Within the introduction the authors should briefly introduce the concept of muscle athropy, the following reference should be useful; Stellavato, A., Abate, L., Vassallo, V., Donniacuo, M., Rinaldi, B., & Schiraldi, C. (2020). An in vitro study to assess the effect of hyaluronan-based gels on muscle-derived cells: Highlighting a new perspective in regenerative medicine. PloS one, 15(8), e0236164. https://doi.org/10.1371/journal.pone.0236164;

Response: We appreciated the reviewer for introducing us to more informative references.

  1. Figure 2B; scale bar are missing, please ass also information about the used magnification;

Response: As suggested by the reviewer, we added information about the magnification and included the scale bars.

  1. Figure 2E, 3B, 4H and 5D; the same of Figure 2B;

Response: As suggested by the reviewer, we added information about the magnification and included the scale bars.

  1. Why did you use housekeeping different for western blotting? Please, specify among the text the reasons;

Response: In the current study, we used GAPDH, HSP90, or b-Actin as a loading control depending on the situation. These proteins have been used as loading controls because they are consistently expressed during myogenic differentiation. When we selected one of them as a loading control, we had to consider the cross-reactivity of antibodies or the difference in molecular weight of the protein during the reblotting process.

- As suggested by the reviewer, we explained this in the Materials and methods section.

  1. Figure 5B; western blotting illustration is not clear;

Response: To make it more recognizable, we put the lines between the samples on the panel of Figure 5B.

  1. It is not clear the origin of used cells, please, give more details;

Response: We originally purchased C2C12 cell lines from ATCC. As suggested by the reviewer, we explained this in the Materials and methods section.

  1. Please, insert specific details about antibodies codes;

Response: As recommended by the reviewer, we added more information on antibodies in the supplementary table.

  1. Please, try to better describe RNA extraction.

Response: As requested by the reviewer, we described the process of RNA extraction in the Materials and methods section.

Reviewer 2 Report

Comments and Suggestions for Authors

The paper by Kim et al. describes the effect of Gloiopeltis tenax aqueous extract (GTAE) in ameliorating muscle atrophy caused by dexamethasone treatment. In particular, the authors first show that GTAE can promote myoblasts differentiation in vitro and has positive effects on mitochondrial biogenesis and function. Next, the protective effect of GTAE on an in vitro model of atrophy (dexamethasone-induced myotubes atrophy) is demonstrated, followed by an in vivo study showing that GTAE administration prior dexamethasone-mediated atrophy induction in mice can alleviate muscle wasting and preserve muscle strength and activity. The paper is overall well written, however the lack of information about compounds found in the Gloiopeltis tenax water extract hinders the understanding of the potential molecular mechanisms involved. I furthermore have the following comments:

1.      As the authors point out correctly in the introduction of the manuscript, sarcopenia is an aging-associated disease usually triggered by decreased physical activity and/or malnutrition in the elderly. Therefore, I wonder why the dexamethasone model of muscle atrophy was chosen for the in vivo study as I feel that other atrophy models, such as hindlimb suspension to mimic physical inactivity or fasting to mimic malnutrition, would have been more relevant. This is of particular importance also since different molecular mechanisms may be in place in causing atrophy upon different triggers, therefore the ameliorating effect of GTAE should be evaluated in different models of atrophy.

2.      There is a complete lack of characterization of the composition of the GTAE extract used in the study. It would be very interesting if the authors could provide some additional analysis aimed at identifying compounds present in the extract. In the discussion (lines 286-292) it seems that different types of extracts were prepared and tested, however the paper does not show results about these preliminary tests. I feel that these results, if included, would add valuable information to the study. Furthermore, were all the experiments performed with the same batch of GTAE extract? In absence of a clear composition of the extract it will be difficult to grant reproducibility of the results obtained in this study. It is also not clear to me how the extract was quantified in order to calculate the experimental doses used in the study.

I furthermore have the following minor/technical comments:

1.      In paragraph 2.1 and figure 1 title the effect of GTAE on PGC-1α is mentioned. However, results shown in figure 1 do not show any information on PGC-1α, as this is instead analyzed in data presented in figure 2.

2.      In figure 1 and figure 3 there is quite a big discrepancy in the levels of Myosin Heavy Chain (MHC) shown in western blot experiments and the levels shown in immunofluorescence images (compare in particular figure 3b and figure 3f, the DEX-induced reduction in MHC shown in the western blot (3f) appears to be much more severe compared to what shown in the IF images (3b)).

3.      In figure 1b it is not clear whether cells were differentiated for 3 days and then treated for additional 24 hours with GTAE. If this was the case, cells that did not receive GTAE look very poorly differentiated, also when compared with differentiated cells shown in figure 3b. The same lack of clarity about the timing of treatments and differentiation is also for results shown in figure 2b and 2d: were cells treated with GTAE at day 0, and then differentiated for 3 days? Was GTAE present in the differentiation medium or was it applied only before the medium switch from GM to DM? If GTAE was applied before the beginning of the differentiation, is there any information about the stability of the compound(s) present in the extracts that may mediate the observed effects?

4.      The central histograms in figure 5c are partially overlapping.

5.      The diagram in figure 3a (Day 3) should be DEX +/- GTAE otherwise it looks like all samples were treated with both drugs.

Comments on the Quality of English Language

The paper is written in good English and clearly understandable. I only detected a couple of typos.

Author Response

  1. As the authors point out correctly in the introduction of the manuscript, sarcopenia is an aging-associated disease usually triggered by decreased physical activity and/or malnutrition in the elderly. Therefore, I wonder why the dexamethasone model of muscle atrophy was chosen for the in vivo study as I feel that other atrophy models, such as hindlimb suspension to mimic physical inactivity or fasting to mimic malnutrition, would have been more relevant. This is of particular importance also since different molecular mechanisms may be in place in causing atrophy upon different triggers, therefore the ameliorating effect of GTAE should be evaluated in different models of atrophy.

Response: Dexamethasone-induced muscle atrophy is a more user-friendly and less invasive approach that mimics sarcopenia, showing similarities to naturally aged mice in muscle mass, function, muscle contraction, and fiber types (J. Orthop. Translat. (2023)39:12-20).

However, the use of this muscle atrophy model as a sarcopenic model has limitations in explaining all aspects of aging. According to the results from other research groups (Cell Rep. (2021)37:109971; Endocrinol. (2017)158:664-77), there are some different molecular changes between aged mice and dexamethasone-treated mice.

- In the current study, we just focused on the verification of the protecting effect of GTAE in muscle atrophy.

- To gain a more detailed understanding of the molecular mechanisms of GTAE in sarcopenia, we fully agree with the reviewer that additional sarcopenic animal models, such as naturally aged mice and progeria mice, are needed.

  1. There is a complete lack of characterization of the composition of the GTAE extract used in the study. It would be very interesting if the authors could provide some additional analysis aimed at identifying compounds present in the extract. In the discussion (lines 286-292) it seems that different types of extracts were prepared and tested, however the paper does not show results about these preliminary tests. I feel that these results, if included, would add valuable information to the study. Furthermore, were all the experiments performed with the same batch of GTAE extract? In absence of a clear composition of the extract it will be difficult to grant reproducibility of the results obtained in this study. It is also not clear to me how the extract was quantified in order to calculate the experimental doses used in the study.

Response: For this study, we initially prepared several extracts of G. tenax through ethanol extraction or water extraction. Three different ethanol extracts were tested for PGC-1α gene expression, but their activation effect was lower than that of the water-soluble GTAE. These results were incorporated into the revised manuscript. Additionally, the analysis of the metabolites of GTAE and the spectroscopic basis for identifying these substances were included in the manuscript and supporting materials.

- For this study, we performed the experiments with two different batches of GTAE. Before starting the experiment, we initially tested the differences in their effects on muscle differentiation, PGC-1a expression, etc. We also considered the difference in the scale of the effects. Both batches produced almost the same effects.

- In the preliminary study, we determined the concentration of GTAE that resulted in the constitutive and reproducible induction of muscle cell differentiation and PGC-1a activation.

I furthermore have the following minor/technical comments:

  1. In paragraph 2.1 and Figure 1 title the effect of GTAE on PGC-1α is mentioned. However, results shown in Figure 1 do not show any information on PGC-1α, as this is instead analyzed in data presented in Figure 2.

Response: As pointed out by the reviewer, we amended the corresponding part in the manuscript.

  1. In Figure 1 and Figure 3 there is quite a big discrepancy in the levels of Myosin Heavy Chain (MHC) shown in western blot experiments and the levels shown in immunofluorescence images (compare in particular Figure 3B and Figure 3F, the DEX-induced reduction in MHC shown in the western blot (3F) appears to be much more severe compared to what shown in the IF images (3B)).

Response: We agree with the reviewer that there is a significant discrepancy in scale between the patterns of MHC fluorescent images and those of MHC Western blots. We can explain that this discrepancy may be due to a difference in tendencies between the visualization of fluorescent dye and the quantification of a protein. The point is that both the results of fluorescent images and the Western blot revealed a similar expression pattern of MHC. Atrophic myotubes or smaller myotubes express a lower level of MHC that was rescued by GTAE. Additionally, we confirmed these data by performing the experiments with three separate sets of cell cultures.

  1. In Figure 1b it is not clear whether cells were differentiated for 3 days and then treated for additional 24 hours with GTAE. If this was the case, cells that did not receive GTAE look very poorly differentiated, also when compared with differentiated cells shown in Figure 3B. The same lack of clarity about the timing of treatments and differentiation is also for results shown in Figure 2B and 2D: were cells treated with GTAE at day 0, and then differentiated for 3 days? Was GTAE present in the differentiation medium or was it applied only before the medium switch from GM to DM? If GTAE was applied before the beginning of the differentiation, is there any information about the stability of the compound(s) present in the extracts that may mediate the observed effects?

Response: The purpose of Figure 1b is to verify the effect of GTAE on myogenic differentiation. Thus, we treated C2C12 cells with GTAE when myogenic differentiation was induced at D0, continued the differentiation, and compared myogenic tube formation at D3. In Figure 2b and 2c, we also treated the cells just before starting differentiation to determine the effect of GTAE on mitochondrial function during myogenesis.

On the other hand, the aim of Figure 3b is to identify the effect of GTAE on muscle atrophy. Therefore, we treated myotube-forming cells with GATE at D3 and monitored the protective effect of GTAE from DEX-induced atrophy for an additional 24hours.

To induce myogenic differentiation with C2C12 cells, we switch the medium from GM to DM at D0 (= cell confluency is almost 80-90%). In the current study, we switched to DM containing DMSO or GTAE for differentiation. In the case of Figure 3b, we replaced DM with DM containing DMSO or GTAE at D3.

  1. The central histograms in Figure 5C are partially overlapping.

Response: As pointed out by the reviewer, we rearranged the panel properly.

  1. The diagram in Figure 3A (Day 3) should be DEX +/-GTAE otherwise it looks like all samples were treated with both drugs.

Response: As suggested by the reviewer, we corrected it to DEX +/- GTAE.

Reviewer 3 Report

Comments and Suggestions for Authors

The present study examines the effect of Gloiopeltis tenax extract on myogenesis and inhibition of muscle atrophy, with interesting results. On the other hand, the present study has serious problems in the experimental method.

The serious problem with this study is that it is unclear whether the extraction method used in this study definitely extracted some bioactive compounds. At present, the possibility cannot be ruled out that the rich nutrients originally contained in G.tenax, such as carbohydrates, proteins, and vitamins, were leached out and the extract solution containing these rich nutrients was simply added to the culture medium. In other words, the concern remains that the experiment was only simply conducted in a nutrient-rich medium, but not any bioactive compounds.

In order to solve this problem, it must be objectively demonstrated that some bioactive compounds in G.tenax can be reliably extracted by the distilled water extraction method used in this study, based on previous studies and data obtained by the present study.

In addition, data comparing the components of each culture medium used and their concentrations should be presented in this paper.

Author Response

Thank you for your comments. To substantiate the points you raised, we have added information about the metabolites of GTAE that were previously omitted from the submission. From the GTAE extract, we isolated 4-hydroxy-N-methylproline, floridoside, and shinorine, and their structures were elucidated using MS and 1H NMR data. Consequently, we have included the spectroscopic data and the process of isolating the metabolites, primarily through hot water extraction, in the supporting material and main text, respectively.

Notably, floridoside is a compound extracted from red algae, and it is known that its extraction yield improves with increased water content during organic solvent extraction (doi: 10.1016/j.carres.2007.07.021) whereas it decreases rapidly during ethanol extraction (doi: 10.1515/BOT.2007.005). In addition, various biological activities of this compound have been reported such as antibacterial, anti-inflammatory and antioxidant properties (doi: 10.3390/md18020105) are well-documented. Based on this, we can infer that the hot water extraction method allows for the extraction of bioactive compounds present in G. tenax rather than merely a nutrient-supplemented culture medium and the rich nutrients originally contained in G. tenax.

- Generally, C2C12 myoblasts are cultured for growth in a nutrient-rich medium, DMEM containing 15% FBS. However, when induced to undergo myogenic differentiation, the medium is replaced by a minimal medium, DMEM containing 2% horse serum, as indicated in the Materials and Methods section. In the current study, we tested the effect of GTAE on myogenesis and muscle atrophy during myogenic differentiation, in other words, under a nutrient-deficient condition. Thus, we can say that the atrophy-protective effect of GTAE is due to a bioactive compound in GTAE, not to a nutrient-rich medium.

- The compositions of growth medium (GM) and differentiation medium (DM) are indicated in the Materials and Methods section.

Reviewer 4 Report

Comments and Suggestions for Authors

The manuscript of Kim et al. discribes a combined in vitro-in vivo study evaluating the muscle function-promoting effects of a red algae extract. The manuscript could be improved by implementing the changes suggested hereunder:

The abstract should more clearly subdivide the results of in vitro/in vivo and specify the in vitro cell line. Also, in vivo results ‘relieving the atrophic effect’ and ‘enhancing mitochondrial function’ are not informative enough and should be given in more detail.

Perform text corrections and remove or rewrite unmeaningful sentences, ex.  ‘Exercise exerts its beneficial effects on muscle strength and function’; Line 35 word missing underlying ‘mechanisms’?, Line 43 ‘mediating by controlling’; ‘Practically’, line 271 ‘special’, line 276 ‘making them regarded as’.

The figure 2 legend mentions qRT PCR for mtDNA/nuDNA ratios. I suppose this should be qPCR? Include this missing method in the M&M section. Also, give more detail on how quantifications of western blots were performed using ImageJ software. Why was HSP90 chosen as a loading control for western blotting? For IF method MF-20 antibody is mentioned (from Cite Ab?), but Table S1 speaks of BA-32 from DSHB. Which is correct, of are these the same clones?

Supplementary files should include full western blots of the panels shown in figures 1, 2 and 3.

In the text, emphasis should be placed on distinction between reduced basal expression and/or reduced induction of  PGC1-a expression. Importantly, in the association with aging.  Also, the known expression regulation mechanisms transcriptional/posttranscriptional should be discussed in more detail.

Line 195 states that muscle strength rescue was a ‘surprise’. Explain or remove.

An important shortcoming of the study is that the GTAE extract was not characterized. The authors also neglect to recognize that an important constituent of G. tenax is water-soluble taurine (doi 10.1007/978-94-024-1079-2_88). Many of the observed effects of GTAE might therefore be attributed to taurine, as the latter is known to positively influence muscle function (doi: 10.3390/metabo12020193), restoring muscle mass through activation of the AKT-mTOR axis (doi: 10.1242/dmm.050540). This relevant literature should be discussed, the (possible) composition of GTAE should be addressed, and the limitation should be acknowledged that the extract is in fact a mixture of varying concentrations of compounds. Make clear comparison between water soluble and hydrophobic (DHA) compounds, and discuss which would be preferred and why.

Do the authors advocate for clinical studies and the development of a recommendation of GTAE in the elderly? Are there any possible counterindications? Is the conclusion section correctly placed (now at the end after M&M section)?

Typo line 148: th effect

Comments on the Quality of English Language

The text needs moderate editing

Author Response

The abstract should more clearly subdivide the results of in vitro/in vivo and specify the in vitro cell line. Also, in vivo results ‘relieving the atrophic effect’ and ‘enhancing mitochondrial function’ are not informative enough and should be given in more detail.

Response: As suggested by the reviewer, we revisited and amended the abstract accordingly.

Perform text corrections and remove or rewrite unmeaningful sentences, ex.  ‘Exercise exerts its beneficial effects on muscle strength and function’; Line 35 word missing underlying ‘mechanisms’?, Line 43 ‘mediating by controlling’; ‘Practically’, line 271 ‘special’, line 276 ‘making them regarded as’.

Response: As suggested by the reviewer, we amended the text section accordingly.

The Figure 2 legend mentions qRT PCR for mtDNA/nuDNA ratios. I suppose this should be qPCR? Include this missing method in the M&M section. Also, give more detail on how quantifications of western blots were performed using ImageJ software. Why was HSP90 chosen as a loading control for western blotting? For IF method MF-20 antibody is mentioned (from Cite Ab?), but Table S1 speaks of BA-32 from DSHB. Which is correct, of are these the same clones?

Response: As the reviewer pointed out, the experiment performed for mtDNA/nuDNA ratio is qPCR, not qRT-PCR. Thus, we described qPCR in the materials and methods section.

- In the current study, we used GAPDH, HSP90, or b-Actin as a loading control depending on the situation. These proteins have been used as loading controls because they are consistently expressed during myogenic differentiation. We also needed to consider the cross-reactivity of antibodies or the difference in molecular weight of the protein during the reblotting process.

- We explained how to quantify and normalize the protein expressions in the Materials and Methods.

- We corrected the information for anti-MHC antibody (MF-20) in the manuscript and the supplementary table.

Supplementary files should include full western blots of the panels shown in Figures 1, 2 and 3.

Response: As suggested by the reviewer, we provided the raw data of the Western blots in the supplementary materials.

In the text, emphasis should be placed on distinction between reduced basal expression and/or reduced induction of PGC1-a expression. Importantly, in the association with aging.  Also, the known expression regulation mechanisms transcriptional/posttranscriptional should be discussed in more detail.

Response: As advised by the reviewer, we clarified the reduction of PGC1-α transcription in association with aging. Also, we explained the post-transcriptional and post-translational modification of PGC-1α in the discussion section.

Line 195 states that muscle strength rescue was a ‘surprise’. Explain or remove.

Response: We agree with the reviewer on this point and removed ‘surprisingly’.

An important shortcoming of the study is that the GTAE extract was not characterized. The authors also neglect to recognize that an important constituent of G. tenax is water-soluble taurine (doi 10.1007/978-94-024-1079-2_88). Many of the observed effects of GTAE might therefore be attributed to taurine, as the latter is known to positively influence muscle function (doi: 10.3390/metabo12020193), restoring muscle mass through activation of the AKT-mTOR axis (doi: 10.1242/dmm.050540). This relevant literature should be discussed, the (possible) composition of GTAE should be addressed, and the limitation should be acknowledged that the extract is in fact a mixture of varying concentrations of compounds. Make clear comparison between water soluble and hydrophobic (DHA) compounds, and discuss which would be preferred and why.

Response: Recently, we were able to isolate a compound from GTAE that demonstrated a positive effect on myogenic differentiation, and we can confirm that this compound is not taurine. However, we would like to ask the reviewers for their understanding that we cannot disclose the specific compound due to a patent issue.

- For this study, we initially prepared several extracts of G. tenax through ethanol extraction or water extraction. Three different ethanol extracts were tested for PGC-1α gene expression, and their activation effect was lower than that of the aqueous extract, GTAE. If we consider that DHA was extracted from the ethanol-soluble extract of G. furcata, it appears that the ethanol extracts of G. tenax in this study either do not contain or contain less DHA.

- As advised by the reviewer, we added a description of the role of taurine in muscle mass and function in the discussion section. Also, we specified that it requires identifying the constituents of GTAE in order to verify the effect of GTAE on muscle atrophy.

Do the authors advocate for clinical studies and the development of a recommendation of GTAE in the elderly? Are there any possible counterindications? Is the conclusion section correctly placed (now at the end after M&M section)?

Response: One of our plans is to develop GTAE as a health functional food or therapy for sarcopenia. Thus, it requires identifying constituents of GTAE and conducting additional scientific research to assess the mode of action of GTAE in muscle atrophy.

- According to the file format instructions, the conclusion section is placed after the materials and methods section.

Typo line 148: th effect

Response: We thank the reviewer for carefully reading our manuscript and for pointing out this spelling mistake. We corrected the misspelling.

Round 2

Reviewer 2 Report

Comments and Suggestions for Authors

I thank the authors for fully addressing my previous comments. I recommend the revised manuscript for publication.

Author Response

Thank you for accepting our revised manuscript.

Reviewer 3 Report

Comments and Suggestions for Authors

I judged that the points pointed out in the previous peer review have generally been improved.

Author Response

(The authors gave the same response as above.)

Reviewer 4 Report

Comments and Suggestions for Authors

The manuscript has been well revised based upon the remarks made. Only two points need to be clarified that concern full western blots:

- In Fig 3, atrogin-1 panel does not seem to match the full blot, with the latter presumably representing a longer (sub-optimal) exposure. Replace with the correct full blot.

- For Fig 1, MHC and b-actin bands need to be marked in the full blots.

Comments on the Quality of English Language

English language is fine.

Author Response

In Fig 3, atrogin-1 panel does not seem to match the full blot, with the latter presumably representing a longer (sub-optimal) exposure. Replace with the correct full blot.

Response: It was a mistake in selecting the raw data for the Western blot. We appreciated the reviewer for letting us recognize it. We replaced the image with the correct one.

For Fig 1, MHC and b-actin bands need to be marked in the full blots.

Response: As pointed out by the reviewer, we provided the raw data of the Western blots in the supplementary materials.